# Cytosine base editing systems with minimized off-target effect and molecular size

Ang Li[1], Hitoshi Mitsunobu[2,3], Shin Yoshioka[1], Takahisa Suzuki[1,4], Akihiko Kondo [1,2] & Keiji Nishida [1,2] ✉

Cytosine base editing enables the installation of specific point mutations without double-strand breaks in DNA and is advantageous for various applications such as gene therapy, but further reduction of off-target risk and development of efficient delivery methods are desired. Here we show structure-based rational engineering of the cytosine base editing system Target-AID to minimize its off-target effect and molecular size. By intensive and careful truncation, DNA-binding domain of its deaminase PmCDA1 is eliminated and additional mutations are introduced to restore enzyme function. The resulting tCDA1EQ is effective in N-terminal fusion (AID-2S) or inlaid architecture (AID-3S) with Cas9, showing minimized RNA-mediated editing and gRNA-dependent/independent DNA off-targets, as assessed in human cells. Combining with the smaller Cas9 ortholog system (SaCas9), a cytosine base editing system is created that is within the size limit of AAV vector.

The cytosine base editing is mediated by cytidine deaminase guided by a nuclease-deficient CRISPR system. At the target site, deamination of cytosine generates uracil, which eventually converts C•G base pair (bp) to a T•A bp without introducing a double DNA strand break[1,2]. The originally developed cytosine base editing systems, Base editor (BE)[1] and Target-AID[2], respectively employ rAPOBEC1 and PmCDA1 as deaminases and efficiently introduce mutations within the editing windows of 12–16 bases and 16–20 bases upstream of the PAM (protospacer-adjacent motif).

Recent studies have raised concern that deaminase-mediated base editing systems can induce genome-wide SNV off-targets especially if overexpressed for long period of time[3,4]. In contrast to CRISPR-Cas9-dependent off-targets which is based on mismatch annealing of guide RNAs, the SNV off-targets induced by base editing appears to be independent of the target sequence and is thought to be caused by non-specific, random deamination by the deaminase domain. BE systems have been well studied and the original BE3 and BE4 have been shown to induce both DNA and RNA off-targets[3–6] and several rAPOBEC1 mutant variants were then identified with reduced off-target effects[5,7,8]. Besides, domain-inlaid type base editors revealed different editing windows and attenuated off-target effects[9–11]. Analytical

methods have also been developed to evaluate off-target potential of the base editors. Genome-wide mutations can be thoroughly elucidated by whole genome sequencing (WGS). However, WGS is expensive, time-consuming, and has low throughput. In addition, it may not be sensitive enough for further comparative analysis of the improved base editors. Previous WGS-based studies had indicated that actively transcribed regions were prone to base editing off-target effect[3,12], because the R-loops which is formed by the exposure of single-stranded DNA by RNA transcription is a preferred substrate for the deaminases. To mimic such hot spots, localized R-loop was formed by using an orthogonal nuclease-defective CRISPR system which was co-transfected with base editing systems targeting another distant locus[7,8]. Deep sequencing of the R-loop region allowed for rapid and sensitive comparative evaluation of gRNA-independent off target potency. Alternatively, the rate of non-specific mutations can be monitored as the occurrence of drug resistant mutants in microbes such as yeast[2] and E.coli[13]. By conducting these studies simultaneously, potential biases can be compensated for each other[7].

For the treatment of genetic diseases, base editing is considered as a promising agent because it can install specific SNV without inducing DNA double-strand breaks or template DNAs. In vivo delivery has

[1]Graduate School of Science, Technology and Innovation, Kobe University, Kobe, Hyogo, Japan. [2]Engineering Biology Research Center, Kobe University, Kobe, Hyogo, Japan. [3]Present address: Bio Palette inc, Kobe, Hyogo, Japan. [4]Present address: Tokyo Metropolitan University, Hachioji, Japan. ✉e-mail: keiji_nishida@people.kobe-u.ac.jp

been one of the major bottlenecks to achieve efficient and specific editing at the target tissue, and smaller molecular size is advantageous especially for in vivo delivery tools such as adeno-associated virus (AAV) vector. AAV vector is one of the promising delivery methods for gene therapy with greater safety and efficiency[14], although its DNA vector size limitation (4-5 kb) hinders wider applications including base editing. For conventional genome editing, smaller CRISPR ortholog *Staphylococcus aureus* (Sa) Cas9 has led to the development of single AAV vector[15,16], but adding a deaminase domain has been challenging[10]. Instead of composing a single AAV vector, the base editing components could be split into two AAV vectors to circumvent the size limitation[17].

In this study, we address structure-based rational engineering of the cytosine base editing system Target -AID to minimize its off-target effects and molecular size, and show the ability of single-AAV mediated cytosine base editing.

## Results

### Elimination of DNA-binding region and restoration of deaminase activity of PmCDA1

DNA deaminases have an intrinsic affinity for DNA and cause non-specific deamination. The structure of hAID, a human homolog of PmCDA1, has revealed its complex formation with double-stranded DNA in a region distinct from the catalytic core[18] (Fig. 1a). Based on the amino acid alignment of hAID and PmCDA1, the potential DNA-binding moieties for PmCDA1 were located to residues 21–27 and 172–192 of the total 208 amino acids (a.a.) length of the protein (Fig. 1a). To delete the predicted DNA-binding region, we first made a series of truncations from the C-terminal end (1–201, 1–197, 1–190, 1–183, 1–179, 1–176, 1–161) and tested their base editing activity in yeast *Saccharomyces cerevisiae* (BY4741) cells (Supplementary Fig. S1). To reflect the changes in their deaminase activity within the dynamic range, we intentionally omitted the uracil DNA glycosylase inhibitor (UGI) which easily increases mutation rate to the saturation level in yeast. Although it has been reported that truncated PmCDA1 (1–161) with UGI showed editing efficiency in yeast comparable to that of full-length PmCDA1 (1–208)[19], our versions which are without UGI and fused to the C-terminus of nCas9 showed a significant decrease in activity as the truncation progressed (Supplementary Fig. S1c). We next performed a series of truncations from the N -terminus of the 1–161 version by fusing to the N-terminus of nCas9. The N-terminus truncations of CDA1 (1–161) first showed further decreased activity, which was then recovered as truncation proceeded to 21 and 28 a.a (Supplementary Fig. S2c). The predicted

structure of the protein indicated that simultaneous truncation of the N- and C- terminus minimizes cross-section and gives a smoother protein surface with less exposure of hydrophobic residues (Supplementary Fig. S2a). Further truncation to CDA1(30–150), which was predicted to be the smallest one with minimum exposure of the hydrophobic surface while retaining its enzymatic core domain intact showed recovered activity (Fig. 1b, Supplementary Fig. S2). These results suggest that the changes in their editing activity were attributed to the conformational stability of the protein. To further improve its activity, we introduced a series of mutations to the hydrophobic residues that were exposed after the truncation. Six mutations were tested in the first round and W122E was found to significantly gain activity to CDA1(30–150) (Supplementary Fig. S3). Additional seven mutations were tested in combination with W122E to find W139R/Q with further improvement of the activity (Supplementary Fig. S3). CDA1(30–150) containing W122E and W139Q was termed as tCDA1EQ hereafter.

As the engineered deaminase supposedly has less affinity to DNA by itself and might be less stable than the original PmCDA1, nCas9-fusion architecture may have a greater impact on its base editing property. Other than fusing to the nCas9 termini, the deaminase can be inlaid in the middle by splitting nCas9 polypeptide and fusing both termini of the protein to the split site. Structurally, 1054 a.a. position in RuvC domain of Cas9 is on the protein surface with flexibility and close to the non-target DNA strand[9] which is subject to deamination. While N-terminally fused tCDA1EQ showed varying editing efficiency among target sites assessed by CAN1 assay[2], the inlaid version showed consistent editing efficiencies comparable to that of the original Target-AID (Supplementary Figs. S4, S5a).

To assess non-specific, gRNA-independent off-target effects, we performed a measurement of the occurrence of thialysine-resistance mutants (LYP1 assay)[2] for the engineered versions fused with UGI. Both N-terminal and inlaid tCDA1EQ versions showed significant decreases (5–79 fold) in the mutant occurrences compared to the original Target-AID (Supplementary Fig. S5b), indicating that their gRNA-independent off-target effects were greatly reduced. We named these N-terminal and inlaid tCDA1EQ versions as AID-2S (Small and Specific) and AID-3S (Small, Specific and Superior), respectively (Fig. 1c).

### Characterization of AID-2S and AID-3S in mammalian cells

Next, we evaluated the editing efficiency and window of AID-2S and AID-3S in human HEK293T cells and compared them with existing improved cytosine base editors YE1, YE2, and R33A + K34A that were reported to exhibit reduced off-target effects[7]. The eight random on-

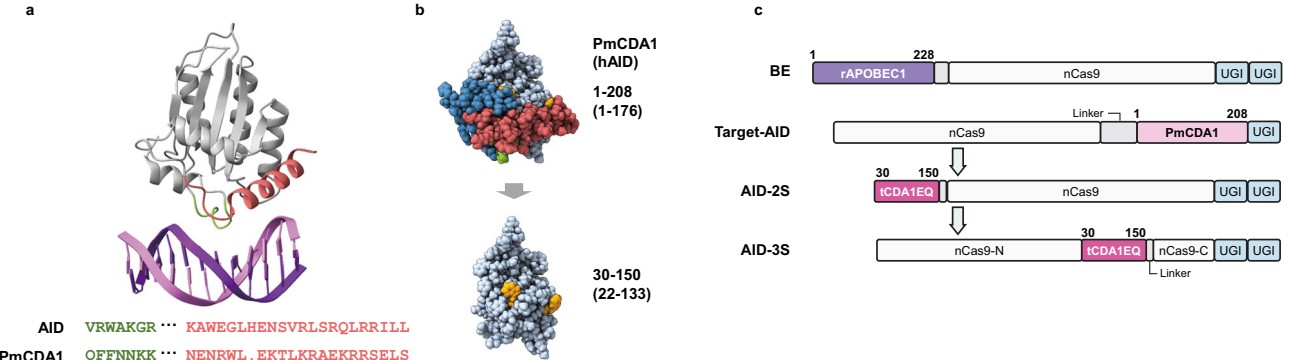

**Fig. 1 | Rational engineering of smaller and specific Target-AID. a** Ribbon model of the structure of a complex of human AID with dsDNA. The non-catalytic dsDNA binding domain is shown in green (N-terminus) and red (C-terminus), of which amino acid sequences are aligned with that of PmCDA1 at the bottom. **b** The predicted space-filling structure of PmCDA1 before and after engineering. In addition

to the direct DNA-binding sites (green and red), segments shown in blue were trimmed to minimize the protein section. The mutated amino acids (W122 and W139) are marked in orange. **c** Domain arrangements of CBE variants used in this study. BE architecture is common to YE1, YE2, and R33A + K34A, except for the point mutations in rAPOBEC1.

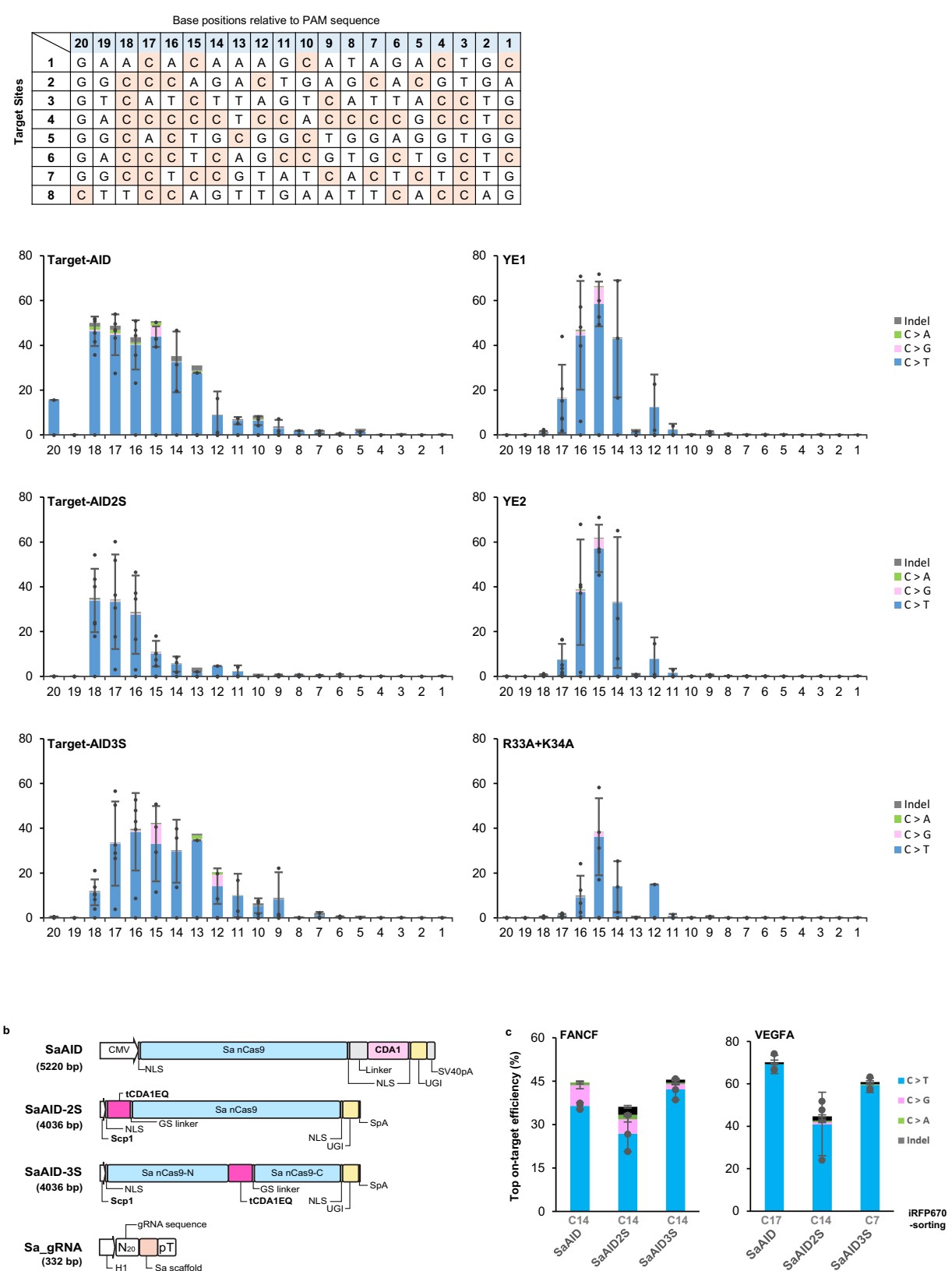

target sites were edited by plasmid DNA vector transfection and analyzed by amplicon deep sequencing. Editing efficiency of cytosine at each nucleotide position relative to the PAM sequence is shown in Supplementary Fig. S6 and averaged in Fig. 2a. Target-AID was capable to perform >20% of editing to all sites. AID-2S, AID-3S, YE1, and YE2 also

performed well except for site 8. R33A + K34A showed target-dependence that obtain >20% of editing only in sites 3, 4, and 7. AID-2S showed a narrower editing region, while AID-3S showed a shift of the editing window to the proximal side of the PAM sequence compared to the original Target-AID.

**Fig. 2 | On-target editing of AID-2S, −3S and rAPOBEC1 base editors in HEK293T. a** On-target editing profiles of the CBE variants analyzed by deep sequencing at eight sites in HEK293T. The sequences of the eight target sites are aligned and the nucleotide positions are numbered from the next to the 5′ end of the PAM sequence. The editing efficiency at each cytosine base position were averaged across all sites. Data are presented as mean values ± sd ($n = 8$). The mutation frequencies for each target sites are also shown in Supplementary Fig 6. **b** Domain architectures of SaAID, SaAID-2S and SaAID-3S. The gRNA expression cassette was combined into each effector plasmid. **c** On-target editing frequencies at the peak base position are shown for of SaAID, SaAID-2S and SaAID-3S, and data are presented as mean values ± sd ($n = 3$). To normalize transfection efficiencies, cells were sorted by the expression of iRFP670 from the plasmid backbone. The mutation frequencies of each nucleotide position are also shown in Supplementary Fig. 11. Source data are provided as a Source Data file.

## Assessment of DNA and RNA off-target effects

Both DNA and RNA off-target effect has been the issue for base editing as well. Intrinsically, APOBEC-based base editors have RNA editing capacity which causes gRNA-independent and gRNA-dependent RNA off-target effects, which facilitated to develop several variants with improved specificity[5,8,20]. To evaluate the new variants, we conducted target RNA sequencing of three mRNA sites (Fig. 3a, CTNNB1, RSL1D1, and IP90) which is reportedly to exhibit high gRNA−independent off-target[8] and six gRNA−dependent mRNA sites (Supplementary Fig. S7, HEK3_1, 2, 3 and RNF2_1, 2, 3)[21]. Both gRNA-independent and gRNA-dependent RNA-editing exhibited by BE4max as reported was significantly lowered by the all other variants tested, while YE1 retained detectable RNA off-target at several sites (Fig. 3a and Supplementary Fig. S7). Original Target-AID and the new variants did not show significant RNA off-target, consistent with the previous report[20].

The gRNA-dependent DNA off-target effect was also assessed by deep-sequencing of the 6 reported sites (HEK2_OT1, 2; VEGFA_OT1, 2, 3, 4)[22,23] (Fig. 3b, Supplementary Fig. S8). AID-2S and AID-3S showed substantially reduced off-target especially in HEK2 site 1 and VEGFA site 2, 3, 4. YE2 and R33A + K34A showed improved off-target editing at all the sites analyzed compared to YE1.

The gRNA-independent DNA off-target effects were assessed by using orthogonal SaCas9 R-loop assay[7] in HEK293T cells. SaCas9 off-targets site 1-6 were selected following the previous studies[7] and an additional site 7 (VEGFA locus) was chosen as its C-rich context may provide higher sensitivity to deamination by CBEs. Target-AID showed detectable off-target editing at all seven sites (Fig. 3c, Supplementary Fig. S9), while AID-2S showed no detectable off-target occurrence in the site 1, 3 and significantly reduced off-target editing at site 2, 5, 6, 7, which was comparable to YE2 and R33A + K34A. YE1 showed rather higher off-target editing at site 6 and 7. AID-3S showed the lowest, hardly detectable off-target editing across all seven sites tested. This may be attributed to the inlaid architecture which sterically limits the access of the enzyme beyond Cas9-binding DNA strand, in addition to the eliminated DNA affinity. On average, AID-2S and -3S respectively exhibited approximately 4.5-folds and 13.7-folds reduction of R-loop off-target editing compared to the original Target-AID while maintaining appreciable on-target editing efficiency (Fig. 3c, d). Combined with yeast LYP1 assay, these consistently support that the genome wide, gRNA-independent off-target effect is greatly mitigated in AID-2S and -3S.

To further define the effect of domain inlaid architecture, inlaid-YE1, inlaid-YE2 and inlaid-R33A + K34A were constructed, resembling AID-3S architecture which splits Cas9 at 1054 a.a. position. Additionally, the 1247 a.a. position of Cas9 which is also predicted as a flexible position[9], was tested as inlaid(1247)-YE1. These rAPOBEC1 inlaid variants, however, showed overall reduction in on-target editing efficiency, especially inlaid-YE2 and inlaid-R33A + K34A by evaluating seven target sites (Fig. 3d, Supplementary Fig. S10). The off-target effect of YE1 was substantially reduced by both inlaid architecture, while no or little improvement was seen for inlaid-YE2 and inlaid-R33A + K34A. (Fig. 3c, Supplementary Fig. S9).

## Minimization of cytosine base editing system and load onto AAV

The engineered PmCDA1 (tCDA1EQ) is substantially smaller (121 a.a.) in size compared to the wild-type (208 a.a.). Smaller molecular size as a genome editing component is advantageous especially for in vivo delivery tools. Therefore, we introduced the small ortholog SaCas9 system to minimize the size of base editing system furthermore (Fig. 2b). To develop SaAID-3S, tCDA1EQ was inlaid into 615−616 a.a. position of nSaCas9 within the HNH domain facing to the polynucleotide-binding cleft. Small Scp1 promoter and SpA terminator were also employed to compose a total length of 4036 bp plus 332 bp of gRNA expression cassette. For comparison, the conventional form of SaCas9 version of Target-AID (SaAID) was also developed, which contains full-length PmCDA1 with linker, UGI, CMV promoter, and SV40 terminator to compose a total length of 5220 bp without gRNA cassette. To normalize transfection efficiency that may vary depending on the vector size, the transfected cells were sorted by the fluorescent signal of iRFP670 expressed from the vector backbone. At the two target sites tested, SaAID and SaAID-3S showed comparable editing efficiency (Fig. 2c) with differences in mutation window (Supplementary Fig. S11a). Relative gRNA-independent DNA off-target effect was assessed for SaAID and SaAID-3S by employing four SpCas9 targets as R-loop off-target sites. SaAID showed significant off-target effect at the four R-loop sites tested, all of which were substantially reduced in SaAID-3S (Supplementary Fig. S11b). The RNA off-target was also assessed at the three sites (CTNNB1, RSL1D1, and IP90), showing no detectable editing (Supplementary Fig. S11c).

To develop a single AAV base-editing vector, a series of viral particles were prepared with varied compositions of AAV-SaAID-2S (A2S-1 to A2S-8) and AAV-SaAID-3S (A3S-1 to A3S-7) by using Scp1, CMV, minCMV or Scp3 promoters, bGH or Short-PolyA terminators and H1 or U6 gRNA promoters. Their editing efficiency was evaluated in vitro by infecting HEK293T cell in comparison with the previously developed dual AAV partitioning system (DUAL-533 and DUAL-739). DUAL-533 is based on v5 *S. aureus* CBE using intein-split SaBE3.9max[24], which had been developed by splitting SaCas9 at 533 a.a. position and ligating via Npu DnaE intein. DUAL-739 is based on Npu DnaE Intein-SaKKH-BE3[25], splitting SaCas9 KKH variant at 739 a.a. position (Supplementary Fig. S12a). DUAL-533 retained the editing efficiency which is consistent with the previous study[24]. Although no significant editing was observed for any of A3S series tested and DUAL-739, we found significant editing efficiency for the A2S series, especially those with smaller compositions (Supplementary Fig. S12c).

## Discussion

In this study, we developed high-fidelity cytosine base editors through structural engineering of the deaminase PmCDA1 to remove the potential non-specific DNA binding moiety. Although the previous studies have explored a series of C-terminal truncations of PmCDA1 to show narrowed editing window and reduced genome-wide off-targets in yeasts[19], simple stepwise removal of the region led to a substantial reduction of its net deaminase activity when measured without UGI. As the UGI-fusion form is so effective in yeast that it may mask the evaluation of the net activity of the deaminase, which may not be readily applicable in other organisms. We intentionally omit UGI for strict evaluation of the enzymatic activity in yeast, then add UGI in the following off-target assay and human cell applications. Based on the predicted structure, we deliberately truncated both N- and C- termini of PmCDA1 to cleanly eliminate the DNA-binding domain and to minimize the protein section. The amino acid substitutions to lessen hydrophobicity at the exposed surface further recovered the activity, probably due to improved protein stability or folding. The obtained

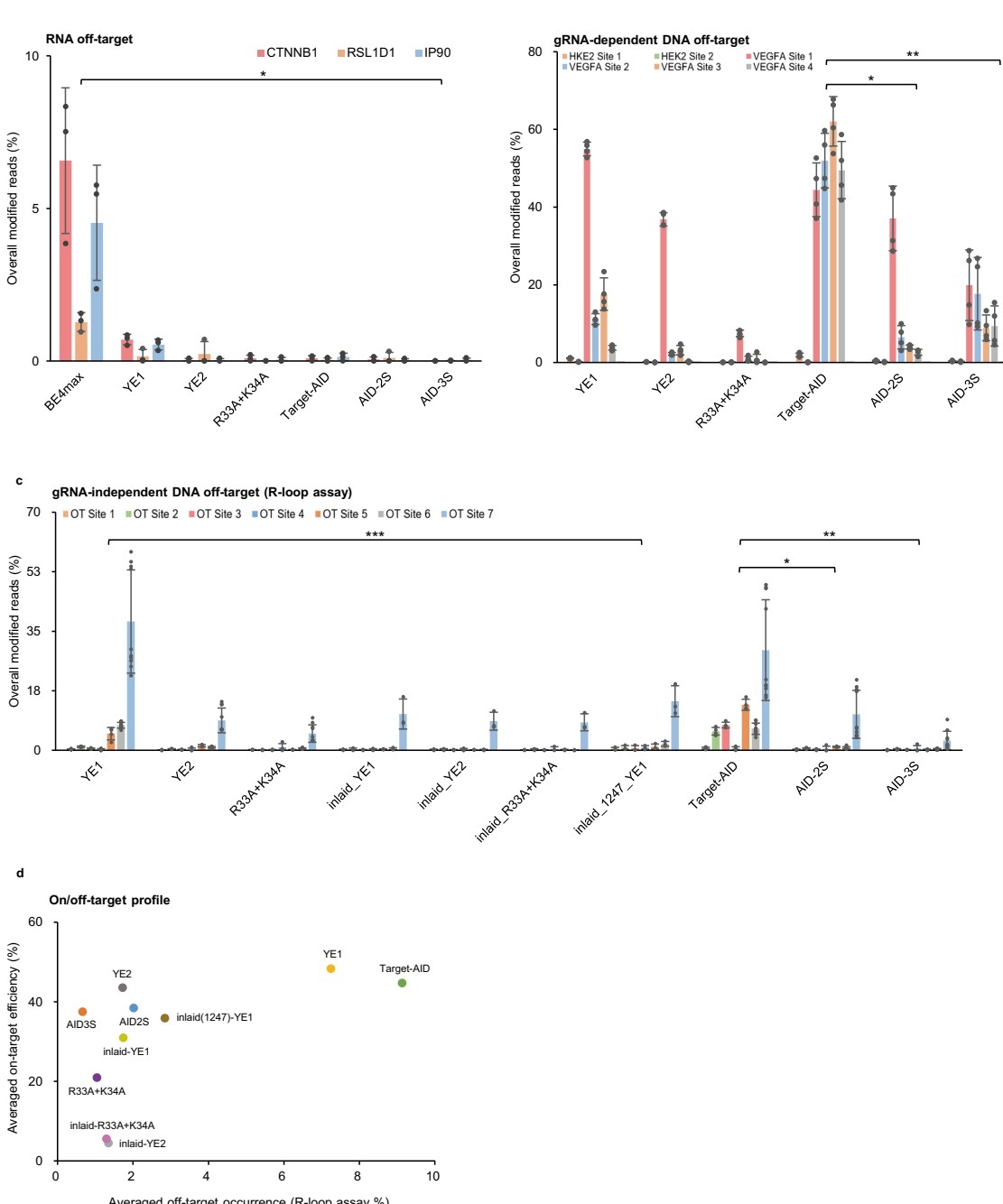

**Fig. 3 | Off-target assessment of AID-2S, −3S and rAPOBEC1 base editors.**
**a** Targeted RNA sequencing of three gRNA-independent locus reported before
(CTNNB1, RSL1D1, and IP90) was performed for four rAPOBEC1 CBEs and three
Target-AID CBEs. Data are presented as mean values ± sd ($n$ = 3). Statistically sig-
nificant differences between BE4max and other CBEs were supported by two-tailed
unpaired Student's t-test; p-value ($P^*$ = 1.98E-03 - 2.36E-03 for RSL1D1 site, 1.45E-02
- 2.18E-02 for IP90 site, 8.90E-03 - 1.33E-02 for CTNNB1 site). The gRNA-dependent
RNA off-target was also shown in Supplementary Fig. S7. **b** Evaluation of Cas9-
dependent off-target effects. The two HEK2 off-target sites and four VEGFA off-
target sites were analyzed by DNA amplicon deep sequencing. AID-2S, -3S showed
significant reduction compared to Target-AID at HEK2 site 1 and VEGFA site 2, 3, and
4, supported by the two-tailed unpaired Student's t-test; p-value ($P^*$ = 1.79E-06 -
9.88E-04 for AID2S, $P^{**}$ = 6.02E-06 - 8.13E-04 for AID3S). The mutation frequencies
at each nucleotide position are shown in Supplementary Fig. 8. Data are presented

as mean values ± sd ($n$ = 3). **c** gRNA-independent DNA off-target was compared by
using seven off-target R-loop sites (see Methods for detail). Compared to Target-
AID, AID-2S and -3S showed significant reduction across all sites, supported by two-
tailed unpaired Student's t-test; p-value ($P^*$ = 3.36E-06 - 5.37E-03 for AID2S,
$P^{**}$ = 2.89E-06 - 8.93E-04 for AID3S). Inlaid 1247_YE1 also showed a significant dif-
ference to YE1 at site 5, 6 and 7 ($P^{***}$ = 4.59E-02, 7.59E-04, 1.59E-04). Off-target
frequencies are shown as the percentage of reads containing the mutation. Data are
presented as mean values ± sd ($n$ > =3). The mutation frequencies at each nucleo-
tide position are also shown in Supplementary Fig. 9. **d** On-target editing versus
average off-target editing profile for all CBEs tested in this study. The y axis
represents the mean on-target editing at the eight on-target, and the x axis repre-
sents the mean off-target editing at the seven orthogonal R-loop sites. Source data
are provided as a Source Data file.

tCDA1EQ version demonstrated comparable on-target activity and greatly reduced off-target effect, especially in the AID-3S inlaid form (Figs. 2a and 3a–d). Furthermore, AID-3S showed a wider editing window and further less R-loop off-target effect than AID-2S. As AID-2S also outperformed pre-existing base editors in on/off-target profile, its narrower editing window should be useful for precise on-target editing.

YE1 was initially developed to narrow editing window[26] then revealed to have reduced off-target effects for both DNA and RNA[27]. While YE2 and R33A + K34A elicited further lowered off-target effects, they performed less robust on-target editing[7]. In this study, YE1 showed relatively high off-target editing, especially at the R-loop site 6 and 7. This might be due to the motif preference of rAPOBEC1 to 5′ TC motif[1,5], which was clearly observed at the both R-loop off-targets (Supplementary Fig. S9). YE2 and R33A + K34A also showed the same trend but much less extent, probably due to weakened substrate binding capacity. In contrast to rAPOBEC1 and other APOBEC family proteins, PmCDA1 apparently did not show such strong motif preferences nor RNA editing[2,20,28]. By exploring whether domain inlaid type will not only affect off-target occurrence in ABE[10,11], but also in rAPOBEC1 CBEs, we introduced inlaid type inlaid-YE1, inlaid-YE2, inlaid-R33A + K34A, and inlaid(1247)-YE1 for comparison. As expected, the inlaid type rAPOBEC1 CBEs showed significant off-target reduction, although their overall on-target efficiency also decreased, possibly due to steric hinderance between Cas9 and rAPOBEC1 whose molecular size is larger with additional domains. Cas9-gRNA dependent DNA off-target effect was also shown to be significantly reduced in AID-2S and -3S (Fig. 3b). Possibly, the DNA affinity provided by deaminase may cooperate with the off-target binding of Cas9-gRNA and subsequent editing.

Minimal off-target effects and robust on-target editing are expected to have a wide range of applications from plant and microbial breeding to clinical use. The AID-3S concept has also been demonstrated with the SaCas9 ortholog, providing the smallest base editing system with appreciable on-target efficiency while maintained the limit off-target effect (Fig. 2c, Supplementary Fig. S11).

In vivo gene therapy requires the delivery of genome editing components to target tissues/cells. Nonviral nanoparticles allow for DNA transport capacities in excess of 10 kbp, but the cytotoxicity and specificity of synthetic polymers are major limitations in their in vivo use. Lentivirus vector has a cargo capacity of up to 8 kb and stable transduction activity, while their genome is randomly integrated into the host chromosome, which poses risks such as inactivation of tumor suppressor genes. Adenovirus vectors carry less risk of genomic insertion and allow for larger cargos, but their application has stalled due to their high immunogenicity. AAV vectors have shown appreciable transduction efficiency and long-term transgene expression in various tissues while maintaining a low level of immunogenicity, although its cargo capacity of the expression cassette is only 4–5 kb. We have developed single AAV vector particle of SaAID-2S that performed substantial editing efficiency. Although it did not outperform the DUAL-533 dual infection at the current version, it is to be noted that the DUAL-533 is based on BE3.9 max that employs high-off-target rAPOBEC1. Single AAV with minimized composition is also advantageous for therapeutic applications because it reduces the preparation process and the number of protein domains that may be responsible for immunogenicity. DAS-739 failed to perform editing, possibly due to target dependence because DUAL-739 employs SaCas9 KKH variant which recognizes altered PAM sequence, or by a technical problem as DUAL-739 C-ter vector exhibited low virus titer. We also did not observe editing activity in A3S series although these are in the same range of size as SaAID2S ones. This may be due to unexpected ssDNA structures or the appearance of sequences that interfered with the function of the AAV vector, because mRNA sequence optimization was done by using full length Cas9 before PmCDA1 was inlaid.

Optimization of the entire sequence to avoid secondary structure and interference, or different order with gRNA cassettes may be tested in the future study. While further exploration is needed for in vivo study, we have demonstrated the ability of a single AAV CBE packaging and editing.

## Methods

### Protein modeling
Prediction of the protein structure of PmCDA1 was done by I-TASSER[29] based on the homology with human AID[18] (PDB: 5W1C).

### Plasmid construction
JM109 chemically competent *E. coli* were used for cloning and preparation of the plasmids by using FastGene Plasmid Mini Kit or NucleoSpin® Plasmid Transfection-grade for transfection. For the yeast experiments, Target-AID variant constructs were made by modifying the original Target-AID vector pRS315e_pGal-nCas9(D10A)-PmCDA1 (Addgene #79617)[2] (Supplementary Data 3). Plasmids for gRNA expression were made from p426-SNR52p-gRNA.CAN1.Y-SUP4t (Addgene #43803)[30] by replacing the target sequence (Supplementary Data 1). For mammalian transfection, plasmid constructs were generated based on YE1-BE4max (Addgene #138155)[7] (Supplementary Data 3). SaCas9-AID versions were made from pX601-AAV-CMV::NLS-SaCas9-NLS-3xHA-bGHpA;U6::BsaI-sgRNA (Addgene #61591)[31]. ScpI promoter[32] and ploy-A tail[33] (Supplementary Data 3) were synthesized by Eurofin. For R-loop assay, nickase SaCas9(D10A) vector was developed from dead SaCas9 (Addgene #138162)[7] by replacing the promoter with ScpI. The U6 promoter gRNA plasmid was constructed based on pU6-Sp-pegRNA-RNF2 (Addgene # 135957). The AAV related plasmid Split-532 (Addgene # 137182, 137183) and Split-739 (Addgene # 119943, 119944. Note: nSaCas9(KKH) arranged to nSaCas9) were introduced for comparison. DNA was PCR-amplified by PrimerSTARMax polymerase (TaKaRa) followed by gel extraction (FastGene Gel/PCR Extraction Kit). Gibson assembly follows the reported protocol[34] and Ligation high Ver.2 (TOYOBO) was used for ligation reactions.

### Yeast experiments
*Saccharomyces cerevisiae* BY4741 cells were transformed by the lithium acetate method and grown in the galactose-induction conditions as described previously[2]. CAN1 on-target mutants and LYP1 off-target mutants were selected by canavanine (60 ug/ml) and thialysine (S-Aminoethyl-l-cysteine) (100 ug/ml), respectively. Mutation frequencies were calculated by colony formation of the serial dilutions on the selection media. The plate images were acquired using an Image Quant LAS 4000 (GE Healthcare Japan, Tokyo, Japan). The on-target site is listed in Supplementary Data 1.

### Mammalian cell experiments
HEK293T cells were cultured in Dulbecco's Modified Eagle's Medium (DMEM) supplemented with 10% fetal bovine serum (FBS) and 1% penicillin-streptomycin. Cells were incubated at 37 °C with 5% carbon dioxide and passaged every 3–4 days. For transfection, cells were seeded onto 48-well poly-lysine-coated plate (Corning®) at the density of 50,000 cells/well with 250 μl of DMEM and incubated for about 24 h. For both on-target and R-loop assay, cells were transfected with 300 ng of base-editor plasmid, 300 ng of nSaCas9 plasmid, 200 ng of SpCas9 guide RNA plasmid, and 200 ng of SaCas9 guide RNA plasmid by FuGENE® HD Transfection Reagent (Promega) following the manufacturer's instructions. For transfection controls, GFP expression plasmid was introduced, while for R-loop assay controls, pUC19 DNA was co-transfected with Sp and Sa guide RNA plasmid together to keep the total quantity of transfected DNA at 1000 ng. The transfected cells were incubated for 24 h and the medium was exchanged with 250 μl of fresh DMEM. The cells were harvested 72 h after transfection and the

genomic DNA was extracted by using Kaneka Easy DNA Extraction Kit (Version 2). Genomic RNA was extracted by using NucleoSpin RNA Plus, and cDNA was generated by using High Capacity cDNA Reverse Transcription Kits (Applied Biosystems).

## Fluorescence-activated cell sorting (iRFP670)
To normalize the transfection efficiency, cell sorting was performed for all-in one SaCas9-AID versions. HEK293T cells were seeded in a collagen-I-coated 24-well plate (IWAKI) at a density of 100,000 cells/well with 500 μl DMEM. Transfection proceeded after ~24 h incubation and a total of 1,000 ng plasmid was applied along with 2 μl of FuGENE® HD Transfection Reagent. The transfected cells were incubated for 24 h and the medium was exchanged with fresh 500 μl DMEM. The cells were harvested 72 h after transfection and washed with 500 μl PBS solution. The cells were trypsinized and resuspended with DMEM media and centrifuged at 160 × g for 2 min to collect the cells. After removing the solution and wash with 800 μl PBS solution, cells were resuspended into fresh 800 μl PBS solution and filtered through cell strainer caps (Falcon®). Finally, cell sorting was executed by SH800S Cell Sorter (Sony Bio-technology Inc., Japan) with the standard iRFP670 expression plasmid setting following the manufacture's instruction. The gating strategy was shown in Supplementary Fig. Flow Cytometry_ FACS iRFP670 gating strategy.

## AAV production and infection
AAV production was performed in HEK293T. The cells were cultured in Dulbecco's Modified Eagle's Medium (DMEM) supplemented with 10% fetal bovine serum (FBS) and 1% penicillin-streptomycin. Cells for production were seeded 3 × 10^6 cell/100 mm dish one day before PEI transfection. 6 μg AAV DNA, 6 μg pHelper, and 6 μg cap plasmid were co-transfected per plate. After 72 h incubation, cells were collected by a rubber cell scraper (Corning). AAV particles were extracted by AAVpro® Extraction Solution (Takara), and AAV purification was processed by Cryonase Cold-Active Nuclease (Takara). After 37 ˚C 30 min incubation, the 1:1 volume of chloroform was added and vortex vigorously for 10 s. Centrifuge 3000 × g for 5 mins and collect the upper water layer. AAV particles concentrated by Amicon® Ultra Centrifugal Filters for 30 mins. The concentrated viral solution was quantified via Droplet Digital PCR (ddPCR) followed ddPCR Supermix for Probe protocol (Bio-Rad), and stored at −30 ˚C. For infection, HEK293T cells were seeded onto 48-well poly-lysine-coated plate (Corning®) at the density of 10,000 cells/well with 250 μl of DMEM one day before. MOI: 2.5 × 10^5 was used in this study. Cells were harvested 72 h after infection and the genomic DNA was extracted by using Kaneka Easy DNA Extraction Kit (Version 2).

## Next generation sequencing (NGS)
The targets used in this study are listed in Supplementary Data 2. Target region-containing fragment was first PCR-amplified using 1st primer pairs from the extracted genomic DNA. The second PCR was performed to obtain adapter added amplicon (~220 bp) by using the first PCR products as template and 2nd primer pairs containing adapter sequences. The amplicon was labeled using NEBNext Multiplex Oligos for Illumina. The DNA sequenced were analyzed by Miniseq system (Illumina, CA, USA) to obtain paired 2 × 150 bp read length by using MiniSeq Mid Output kit (300 cycle). Obtained Fastq data were processed and analyzed by Crispresso2[35]. All analyzed data were obtained with paired end reads, and Fastq sequence data were deposited to the NCBI Sequence Read Archive (PRJNA721271).

## Statistics & reproducibility
Sample size was determined based on pervious reported study[2,7]. Sufficient sample number were chosen to perform statistics test and statistically significant differences between means ($P < 0.05$) were determined using two-tailed unpaired Student's t-test. Data are presented as mean values ± sd. For, randomization, when seeding cell into separate well plate, experiment group and control group are assigned randomly for each replicates. Experiments are systematically designed to treat each sample the same way at the same time. No data were excluded from the analyses.

## Reporting summary
Further information on research design is available in the Nature Research Reporting Summary linked to this article.

## Data availability
The raw DNA sequencing data generated in this study have been deposited in the NCBI Sequence Read Archive database project number PRJNA721271. The data corresponding for each figure are published in the Source Data file. Plasmids for yeast and mammalian expression of Target-AID2S and Target-AID3S have been deposited to Addgene for distribution. Yeast_Target-AID3S (Addgene No. #188646), Yeast_Target-AID2S (Addgene No. #188647), Mammalian_SpCas9_Target-AID2S (Addgene No. #188648), Mammalian_SpCas9_Target-AID3S (Addgene No. #188649), Mammalian_SaAID3S (Addgene No. # 188651), Mammalian_SaAID2S (Addgene No. # 188652), Mammalian_SaAID (Addgene No. # 188653), AAV_A2S-8 (Addgene No. # 188654). The sequence information is listed in Supplementary Data 3. Source data are provided with this paper.

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

## Acknowledgements

The following plasmids were obtained via Addgene. YE1-BE4max (Plasmid #138155), dead SaCas9 (Plasmid #138162), Cbh_v5 AAV-saCBE N-terminal (Plasmid #137182) and Cbh_v5 AAV-saCBE C-terminal (Plasmid #137183) were kindly deposited by David Liu's lab. pLV302 (Plasmid #119943) and pLV312.3 (Plasmid #119944) were kindly deposited by Gerald Schwank lab. pX601-AAV-CMV::NLS-SaCas9-NLS-3xHA-bGHpA;U6::BsaI-sgRNA (Plasmid #61591) was kindly deposited by Feng Zhang lab. This work was supported by the New Energy and Industrial Technology Development Organization (NEDO) to K.N., the Japan Agency for Medical Research and Development (AMED) under Grant Number 20dm0207001 and 21ek0109448h0002 to K.N., Program on Open Innovation Platform with Enterprises, Research Institute and Academia (OPERA) to K.N. and Program for Building Regional Innovation Ecosystems from the Ministry of Education, Culture, Sports and Technology (MEXT) of Japan and the JSPS KAKENHI (Grant number 26119710, 16K14654) to K.N.

## Author contributions

A.L. conceived and designed experiments with input from H.M., S.Y., T.K., and K.N. A.L. performed plasmid construction, yeast assay, human cell assay, and NGS with assistance from H.M. and S.Y.; A.L. performed cell sorting with assistance from T.K. and S.Y.; K.N. and A.K. supervised the entire project. H.M. and K.N. wrote the manuscript and all authors edited the manuscript.

## Competing interests

A.L., K.N., and H.M. are inventors on a patent filed by Kobe University (PCT/JP2020/149419 on PCT international application). The patent application covers the method and the complex of base editing using truncated and mutated deaminase with lowered off-target and molecular size. K.N. and A.K. are members of the board of BioPalette, a genome editing company, and hold shares in the company. H.M. is currently an employee of BioPalette. Other authors declare no competing interests.
