## [Peer Review File · Nature Communications]

Reviewers' Comments:

Reviewer #1:

Remarks to the Author:

In this revised manuscript, the authors well addressed most of points raised by the reviewers. The manuscript has been substantially improved. Nevertheless, I would like to see the following points to be addressed in the manuscript before its publication in Nature Communication.

Comments/Suggestions,

1. Given AID-3S didn't show editing activity in HEK293T using AAV delivery, I suggest the author to recompose or delete their statements such as "we have demonstrated the ability of a single AAV CBE system with minimized size and off-target" in line 253-254, and discuss more about why AID-3S didn't work using AAV delivery and how to further optimize, instead blame the ssDNA structure.
2. It should be "A3S-7" in line 184.

Point-by-point responses to reviewers and editorial points.

We thank again for the critical review and helpful comments by the referee. Our point-by-point responses are listed as follows.

New comments from Reviewer #1

Given AID-3S didn't show editing activity in HEK293T using AAV delivery, I suggest the author to recompose or delete their statements such as "we have demonstrated the ability of a single AAV CBE system with minimized size and off-target" in line 253-254, and discuss more about why AID-3S didn't work using AAV delivery and how to further optimize, instead blame the ssDNA structure.

Responds: Thank you for the comment. We rewrote lines 254-260 and discuss more on the problem of AAV-AID-3S with possible solution.

It should be "A3S-7" in line 184.

Responds: We apologize for incurrent information, we changed into A3S-7.